# Waste Management in a Sustainable Circular Economy as a Part of Design of Construction

**Marcela Spišáková, Tomáš Mandičák ***, **Peter Mésároš and Matej Špak**

Faculty of Civil Engineering, Technical University of Košice, Vysokoškolská 4, 042 00 Košice, Slovakia;
marcela.spisakova@tuke.sk (M.S.); peter.mesaros@tuke.sk (P.M.); matej.spak@tuke.sk (M.Š.)
* Correspondence: tomas.mandicak@tuke.sk; Tel.: +421-55-602-4378

**Abstract:** The Architecture, Engineering, and Construction (AEC) industries are the producers of the most significant waste stream in the European Union. Known EU initiatives propose to deal with the issue of construction and demolition waste (CDW) according to the principles of a circular economy: the 3Rs (reduce, reuse, and recycle). CDW is generated during the whole life cycle of construction. The lack of information about the quantity of CDW during the design phase of building needed for sustainable design of construction was identified as a research gap. The aim of our research is to quantify construction and demolition waste during the construction design phase in a circular economy. The proposed method is based on the generation rate calculation method. This paper describes the proposed methodology for quantifying selected types of construction waste: excavated soil, concrete, and masonry. This information is essential from the point of view of a sustainable circular economy. The main contributions of the paper were identified during the decision-making process of sustainable building design, during the audit of CDW management, and during building information modelling as a support tool for CDW management. As early as the construction design phase, there is the possibility of choosing technologies, construction processes, and materials that have a higher degree of circularity in the economy.

**Keywords:** waste management; sustainable circular economy; circular economy; construction and demolition waste; quantifying of construction and demolition waste; construction projects

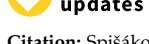



## 1. Introduction

The circular economy (CE) is currently an integral part of sustainability [1]. There are many definitions of circular economics and descriptions of CE and sustainability interactions [2]. The idea of a CE was determined in the late 1970s [3]. Many researchers have influenced the present understanding of a CE's features in the industry, having dealt with the industrial ecology [4], the cradle to cradle of a low cycle assessment [5], the modularity [6], the governance [7], and the closed-loop model [8]. The most used definition of a CE is presented by the Ellen MacArthur Foundation, where a CE is defined as "an industrial economy that is restorative or regenerative by intention and design" [9]. From a supply chain point of view, a CE represents a closed-loop material flow [10]. A CE, from the point of utility value, is defined by Webster [11] as follows: "The circular economy is restorative by design, and which aims to keep products, components and materials at their highest utility and value, at all times". The circular model is also intended to ensure a healthy environment. Profit within this system is based on the efficient use of natural resources, which mainly includes products and/or their components, as well as materials. This significantly minimizes waste and costs for the input materials and energy required to produce new products.

A linear economy is a set of processes for extracting, producing, and disposing of often natural, non-renewable raw materials [6]. A CE is an economic model based on the (repeated) return of materials, components, and products back to the production process [12].

Their circulation radically minimizes waste, such as the energy consumption otherwise required to produce new inputs and total production costs. The benefits of applying the circular economy are also supported by the ideas of the sustainable manufacturing and industry 4.0 platform [13]. CEs presented as 9.1% in the year 2019 and as 8.6% in the year 2020 in the world [14]. Stahel and Reday [8] determined the features of the circular economy within the industrial economy. The current understanding of a CE lies in the idea of a closed loop. The application of the closed-loop principle represents a transition between the CE and sustainability in an industrial economy [15]. A circular economy, in contrast to a linear economy, favors the environmental and social aspects of a sustainable industry system [16] and presents an alternative to a linear economy. A CE uses the idea of a closed loop in supply chains. According to Wang and Wang [17], a closed loop "involves the movement of the products from the upstream suppliers to the downstream customers and the flow of used ones back to the remanufacturers, combine the forward logistics with the reverse logistics". A closed loop allows for unused materials or waste to be returned to the supply chain as recycled raw materials.

A supply chain is characterized by several definitions. The widely used term, as given by Christopher, is "the network of organizations that are involved, through upstream and downstream linkages, in the different processes and activities that produce value in the form of products and services in the hands of the ultimate customer" [18]. Authors Akintoye et al. [19] determined the upstream and downstream in the Architecture, Engineering, and Construction (AEC) industries in terms of supply chains. The upstream in the AEC industries are the suppliers and subcontractors, and the contractors and material manufacturers present the upstream in the supply chain. Thunberg et al. has argued that the main aims of supply chain management are as follows: (i) achieving efficiency in material flow, (ii) achieving efficiency in communication, and (iii) project complexity [20]. The understanding of the effects of CE and sustainable aspects on the supply chain is referred as green supply chain management, sustainable supply chain management, closed-loop supply chain management [21], or a circular supply chain [22]. The AEC industries have huge potential for applying the supply chain according to the principles of a CE and sustainability. The circular supply chain's application in the AEC industries makes it possible to reduce the amount of construction and demolition waste and, at the same time, reduce the consumption of raw material. Construction waste will be returned to the supply chain by a closed loop.

## 2. Theoretical Background

The construction does not only present benefits for the environment and society; it does have several negative influences. One of the elements that negatively affects the environment is waste origin.

### 2.1. Construction and Demolition Waste

In total, up to 36% of waste in Europe is accounted for by the construction industry through construction and demolition work. In the USA, this is up to 60%, and, in China, surveys point to an inlay from 30% to 40% [23–26]. The generation of construction waste, as well as demolition waste, is related to several construction activities. Among these activities, we can include classic construction work. This includes maintenance and remediation work as well.

In generally, an integral part of CDW issues is CDW management which consist of waste generation, waste collection, waste storage, waste processing and waste disposal [27–29]. Construction and demolition wastes are generated throughout the life cycle of building. We know four basic groups of CDW depending on which phase of the construction life cycle waste is generated: wastes from the operation of construction site facilities; wastes from the construction activities containing also preparatory, ancillary and transport of processes; waste from the use of the construction; wastes from the renovation, modernization and demolition of buildings [30,31]. The waste management is determined by the

waste management hierarchy [32]. CDW disposal is divided into five levels: prevention, reuse, recycling, recovery, disposal. The waste management hierarchy sets out the order of priority of the best environmental choice in the field of waste policy overall. The waste prevention is the most efficient and sustainable strategy for the use of natural resources. Waste reuse is a process that involves separating part of the waste suitable for further direct reuse. This waste may be used for the same activity from which it was generated or in another activity, without changing the properties or form of this waste. Increased attention needs to be paid to this method of waste disposal, especially in construction, given that there is a high potential for applying this principle. An example of this procedure in construction is the reuse of brick products, roofing, etc. The recycling presents a process of reusing previously used materials and products, prevents wastage of resources, reduces consumption of natural materials, reduces the amount of waste stored and reduces energy consumption, thus contributing to the reduction of greenhouse gas emissions against the use of primary materials. The recovery and disposal represent the least appropriate ways of CDW hierarchy from point of the sustainability, e.g. landfilling has a negative impact on environment (water pollution, greenhouse gas emission, global warming etc.) [27–29,33]. The waste management also includes the collection, storage and transport of CDW to the site if its processing. The waste processing includes at least pre-sorting, crushing or grinding, magnetic separation of metal elements and sorting.

The waste produced consists of many types of materials, such as masonry and concrete. This is the largest component of waste. Asphalt, as well as some metals and plastics, can also be included among these wastes. There are also materials with higher recyclability, such as wood or excavated soil [8]. The problem is that this waste also contains hazardous substances (including asbestos, PCBs, and others) [8]. This poses a high risk to human health and to the environment. This is even truer if this waste is not treated in accordance with safety regulations. CDW has several uses. Its most common use is as a substitute for natural material in road construction. This material is also used for landscaping or backfilling [33]. As already mentioned, the most common types of waste are used for various purposes, and these wastes can include mixed concrete waste. In many cases, masonry waste is just as numerous [34].

CDW can also be used to replace materials and raw materials. Some wastes, such as masonry and concrete waste, have a massive potential for recycling. Despite this, this material's information is most often used for backfilling and is unflattering; however, this material can also be used in other ways, such as a crushed material in unbound layers on roads and concrete production. It has excellent drainage properties. This waste can be recycled at an estimated 80%. Additionally, there is information that 25% will be recovered from new concrete [34–36]. The regenerated asphalt can be reused for the purposes of the new mixture. Alternatively, it can also be used as an aggregate in unbound layers of roads and infrastructure. The question of the recycling rate is appropriate for this material; exact statistics do not yet exist, but the rate is estimated to be 10% to 20%. Other materials, such as gypsum which can be used as a gypsum powder in the production of several building materials, do not represent a large amount of waste. The most famous is plasterboard, which can be used in the production of cement. It is also important to note that this can also be used in other sectors, namely agriculture [35,36].

Based on the above information, it can be stated, as confirmed by research, that a large amount of CDW can be recruited and reintroduced into the reproductive process [37]; however, this is not the case in all countries of the European Union (EU). The statistics from 2018 are interesting for the EU countries, which show that the CDW rate is 90% [38]. This does not apply to all countries, and there are differences between these results. Several countries, such as the Netherlands, Ireland, and Malta, can be included among the countries that use reclaimed material very intensively. Other countries have also seen a significant share of recycling, such as Hungary, Lithuania, Italy, and Luxembourg. Countrries such as Estonia, Germany, Portugal, the Czech Republic, Austria, and the United Kingdom dominate this statistic.



Conversely, there are countries with relatively low reuse intensity rates. These are countries such as Slovakia or Bulgaria. They recorded only 24% or 51% of the recycling of these materials, respectively [38].

The ambitious target of up to 70% recycling and recovery rates for these materials under the Framework Directive by 2020 has been met in advance by several EU countries in 2018 [32]. This was based on the document "EU Protocol on Construction and Demolition Waste Management", created in 2016 [39]. The opportunity to achieve this has been defined in several ways. First and foremost, there is the opportunity to identify better resources that can be recycled and reused in the construction sector. Another possibility is to improve logistics, which could significantly help sustainability from two perspectives: is the first is to reduce the carbon footprint and contribute to a better situation. The second is to make better use of these materials. The third view on achieving these values is to improve waste treatment, which would require the legislative framework and political will to address these issues and challenges [39]. Instead, barriers prevent the maximum use of these recycled materials and the achievement of a passive balance [30].

*2.2. Life Cycle of Building and Construction Material*

The life cycle of building and construction materials are closely linked to the strategy of the circular economy. The life cycle of building is divided into four main phases: the (i) design phase, (ii) manufacturing and construction phase, (iii) operation and maintenance phase (use), and (iv) end of life (demolition) [40]. The connection between the life cycle of building and the life cycle of construction materials (Figure 1) are interconnected and interactive. Construction design, choice of construction methods, and technologies affect the consumption of given building materials [41] and, consequently, the preference for the location of construction, namely on-site or off-site. We currently know of many construction approaches that allow the construction of off-site, e.g., modern methods of construction, prefabrication, etc. [42]. The renovation or modernization of a building may occur during its use phase. These activities require the supply of building materials but also create CDW. The phase of construction demolition presents the end of a material's life cycle [43]. The end of the life cycle of a construction material is conditioned by waste management. A directive from the European Union [32] determines waste management procedures and waste disposal methods. Considering the application of a CE and a sustainable supply chain, it is necessary to prefer the principles of the 3Rs (reduce, reuse, and recycle) over landfilling. The demolition of a building does not end the life cycle of a building material, but its life cycle begins through loop-closing [44,45].

Waste management decisions need to be supported by information and communication technologies (ICT). Currently, many approaches provide useful drivers for waste management within the circular economy. Drivers consists of: (i) building information modelling (BIM), (ii) internet of things (IoT), (iii) predictive data analytics, (iv) logistics network optimization [40], (v) geographic information system (GIS) [46], (vi) radio frequency identification (RFID) [47] and (vii) material flow analysis [47]. Waste management can be influenced at every phase of the construction life cycle. The initial decision is in the design phase of construction [48]. The design of the building should be sustainable. A sustainable construction design (also called circular design) should also provide the designer with information on waste generation and its 3Rs options—reduce, reuse, recycle [49]. The decisive knowledge of the generated construction waste is: (i) type, (ii) quantity, and (iii) method for waste disposal. The type of and method for waste disposal are defined by legislation [50]. The crucial point of the decision-making process is the quantifying of construction and demolition waste.

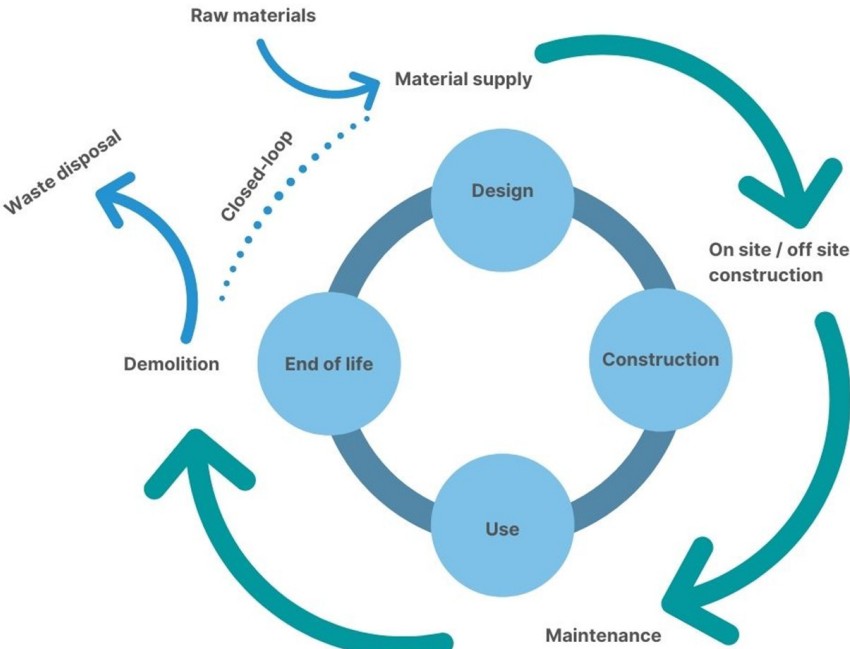

**Figure 1.** Life cycle of building and life cycle of construction materials vs. circular economy and supply chain.

### 2.3. Quantifying of Construction and Demolition Waste

A prerequisite for the implementation of successful waste management is the quantifying of construction and demolition waste (CDW) [51]. Many methods for the quantifying of CDW are known. CDW is generated from different construction activities. The primary activities that generate CDW are:

- Construction of new buildings [52–56];
- Demolition of buildings [56,57];
- Civil and infrastructural works [58].

Methods for the quantifying of CDW may have a general use or use for a specific project. Very essential methods are intended for the project estimation level [52]. An effective national CDW management policy can be supported by the methods for quantifying CDW on the regional estimation level [59].

Authors Wu et al. [52] have identified and classified 57 methods for quantifying CDW. They have processed a three-level classification according to the waste generation activities, estimation level, and particular methodology. The methodology principles determine the level of use. The methods were divided into five groups based on the approach to quantifying CDW:

- Site visit method: direct measurement [60], indirect measurement [53,61,62];
- Generation rate calculation method: per capita multiplier [63,64], financial value extrapolation [64], area-based calculation [65,66];
- Lifetime analysis method: building lifetime analysis [53,56], material lifetime analysis [67];
- Classification system accumulation methods [55,65];
- Variables modelling method [64].

The accuracy and clarity of each method varies. The methods require different input data, which are not always available, and therefore, their relevance cannot be verified. Limitations of quantifying CDW are mainly a lack of actual data and a lack of verification.

The CDW quantification methods can be classified according to their construction activities and estimation levels (Table 1).

**Table 1.** Classification of CDW quantification methods according to construction activities and estimation levels.

| CDW Quantification Method | Construction Activity | | | Estimation Level | | Source |
|---|---|---|---|---|---|---|
| | Construction of New Buildings | Demolition of Buildings | Civil and Infrastructural Works | Project Level | Region Level | |
| DSVM | X | | | x | | [60] |
| ISVM | X | x | | x | | [53] |
| ISVM | X | x | | | x | [61] |
| ISVM | X | x | | x | x | [62] |
| GRC–PC | X | x | | | x | [63] |
| GRC–PC | X | | | | x | [64] |
| GRC–FVE | X | | | | x | [64] |
| GRC–ABC | X | x | | | x | [65] |
| GRC–ABC | X | x | | | x | [66] |
| LAM–BLA | X | x | | x | | [53] |
| LAM–BLA | X | | | x | | [56] |
| LAM–MLA | X | x | | | x | [67] |
| CSAM | X | | | x | | [55] |
| CSAM | X | x | | | x | [65] |
| VMM | X | | | | x | [64] |

DSVM: direct site visit method; ISVM: indirect site visit method; GRC–PC: generation rate calculation method—per capita; GRC–FVE: generation rate calculation method—financial value extrapolation; GRC–ABC: generation rate calculation method—area-based calculation; LAM–BLA: lifetime analysis method—building lifetime analysis; LAM–MLA: lifetime analysis method—material lifetime analysis; CSAM: classification system accumulation method; VMM: variables modelling methods.

Site visit quantification methods are based on a visit to a specific construction site where construction or demolition waste is generated [53,60,61]. The direct method determines the amount of waste based on the spatial dimensions of a place's stockpiled CDW. The indirect method consists of counting the number of trucks for waste transport and the volume of each container.

The generation rate calculation methods are the most used methods for CDW quantification. The principle of this methodology is to obtain the rate of waste generation (coefficient) per specific unit (e.g., $kg/m^2$, $m^3/m^2$) by a different type of multiplier [63–66]. It has a general use; one critique of this method is the absence of suitable databases for the rate calculation.

The lifetime analysis method is used for demolition waste quantification. A building's lifetime analysis considers the age of the building, which often determines the need for demolition work [53,56]. National material flow databases are essential for applying the material lifetime analysis method. The database determines the amount of purchased building material for a given region [67]. The lifetime analysis methods are not generally applicable to every construction project and region. This methodology has researched and designed these databases for the United States only.

The classification system accumulation method comes from the idea of generation rate calculation methods. It is focused on the CDW quantification for a specific waste type and the specific building structure of an element [55,65].

The variables modelling method consists of modelling various variables that arise during construction or demolition activities. These variables include construction time, construction site conditions, weather, material storage, or labor experience. The importance and frequency of these variables affect the generation of construction waste [64]. Authors [66–76] have determined support ICT tools for the quantifying of CDW. The most important approach is building information modelling (BIM), which allows for the extraction of information regarding waste management [68], estimation of the recoverability and recyclability of CDW [69], and algorithms from CDW management [70]. The geographic information system (GIS) is another ICT tool that enables the integration of CDW quantification into a life cycle assessment [71,72]. The GIS enables determination of the location

of waste, which is important from the point of view of the supply chain [73,74]. Radio frequency identification (RFID) is an important ICT tool for the supply chain. It also allows for a smart waste management system based on RFID tracking, scheduling, and handling of waste [75,76].

## 3. Materials and Methods

The aim of our research is the quantification of construction and demolition waste in the construction design phase in the context of a circular economy (Figure 2).

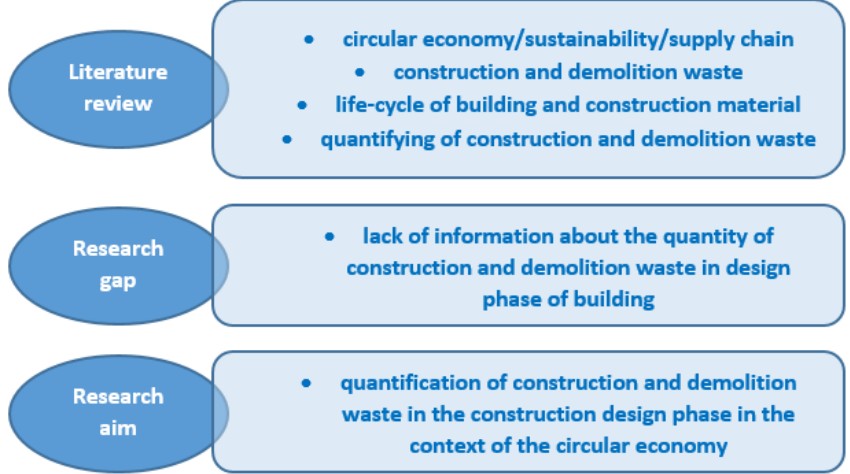

**Figure 2.** Research aim determination.

The research aim was determined based on a detailed literature review. The literature review focused on four main research areas: circular economy, sustainability, and supply chain; construction and demolition waste; the life cycle of building and construction material; and the quantifying of CDW. The detailed literature review found a research gap. Determining a method for the quantification of CDW represents a gap in the research at the regional level. There are no relevant data on this issue in Slovakia [77]; at the same time, there is a lack of a comprehensive information based on the amount of CDW, which would provide relevant data already during the construction design phase [30,77]. Construction design is a crucial phase in the decision-making process for increasing the reduce, reuse, and recycle rate in accordance with the target of the European Union [32]. The methodology proposed for the quantification of CDW in the construction design phase in the circular economy context represents the study's main contribution.

### 3.1. Material of Construction and Demolition Waste Quantification

The material for this research problem is construction and demolition waste. It was necessary to determine the type of construction waste that needed to be quantified within the research. The selection criteria were: (i) the material stream size and (ii) the waste source, which is the most important producer of CDW.

The significant CDW streams, from a material point of view, are concrete, masonry, metals, timber, asphalt, gypsum, glass, and plastics. The statistics (Table 2) confirm that up to 85% of the waste is concrete, ceramics, and masonry [69,78]. The most important sources of CDW are demolition, packaging, and debris [79].

**Table 2.** Construction and demolition waste composition (stream) [78,80].

| Waste Category | %, Min–Max Range |
|---|---|
| Concrete and masonry | 40–84 |
| Concrete | 12–40 |
| Masonry | 8–54 |

**Table 2.** *Cont.*

| Waste Category | %, Min–Max Range |
|---|---|
| Asphalt | 4–26 |
| Others (mineral) | 2–9 |
| Wood | 2–4 |
| Metal | 0.2–4 |
| Gypsum | 0.2–0.4 |
| Plastics | 0.1–2 |
| Miscellaneous | 2–36 |

Based on the information below, the research material is determined as the most important material stream of CDW for demolition work, debris, and packaging, as follows:

- Excavated soil;
- Concrete;
- Reinforced concrete;
- Masonry.

### 3.2. Methods for Quantification of Construction and Demolition Waste

The proposal of a method for the quantification of CDW was preceded by a detailed examination of the existing methods, an analysis of the ICT tools used in Slovakia, and the need to determine the quantification of selected types of CDW. The source of CDW is construction and demolition work. Debris and packaging waste come primarily from construction waste, and demolished waste comes from demolition works [81]. The amount of this waste will be determined as $VADeb_i$apparent volume of debris waste, $VAP_i$—apparent volume of packaging waste, and $VADem_i$ —apparent volume of demolished waste. The input for calculating these volumes is an apparent constructed volume $VAC_i$, which is defined as the volume in cubic meters per square meter build for an item "*i*-th". The items present similar construction elements to those determined from the quantity's take-off, which should be processed before construction begins for each item. Table 3 provides a register of the construction and demolition waste, the code of CDW (according to the European waste catalogue [51,81]), and the unit of measure. The unit of measure for CDW is derived from the unit of measure of the construction element which is the source of an individual CDW. At the same time, the unit of measure (m, m$^2$, m$^3$, kg, pcs) is derived from the unit of measure of the relevant item in the construction and economic software Cenkros 4 [82], which is the most widespread software for processing budgets in Slovakia.

**Table 3.** Register of selected construction activities and relevant construction and demolition waste.

| Construction Activity | Construction and Demolition Waste (CDW) | Code of CDW | Unit of Measure (UM) |
|---|---|---|---|
| Excavation work | Transported excavated soil | 17 05 04 | m$^3$ |
| Concrete work | Concrete | 17 01 01 | m$^3$ |
| Reinforced | Concrete | 17 01 01 | m$^3$ |
| concrete works | Steel (concrete reinforcement) | 17 04 05 | Kg |
| Masonry | Bricks | 17 01 02 | m$^3$ |

The quantification of construction and demolition waste for all types of waste is calculated through an apparent constructed volume of *i*-th items $VAC_i$, which is expressed by the following equation:

$$VAC_i = Q_i * C_i \quad \left[\text{m}^3/\text{m}^2\right] \tag{1}$$

where $VAC_i$ is the apparent constructed volume of $i$-th items [m$^3$/m$^2$]; $Q_i$ is the quantity of $i$-th items [m, m$^2$, m$^3$, t, kg, pcs/m$^2$]; and $C_i$ is the conversion of the amount of the $i$-th item in VAC in m$^3$/Qi specific unit.

The implementation of new building relates to the debris and packaging waste origin (Equations (2) and (3)). Waste volumes are derived and modified from $VAC_i$ using a quantity coefficient of the given activities (debris and packaging).

The apparent volume of debris waste is expressed by the following equation:

$$VADeb_i = VAC_i * C_{Deb_i} = Q_i * C_i * C_{Deb_i} \qquad \left[\text{m}^3/\text{m}^2\right] \tag{2}$$

where $VADeb_i$ is the apparent volume of debris waste of of i-th items [m$^3$/m$^2$]; $VAC_i$ is the apparent constructed volume of i-th items [m$^3$/m$^2$]; $C_{Deb_i}$ is the transformation coefficient of debris waste from $VAC_i$ to $VADeb_i$ [dimensionless]; $Q_i$ is the quantity of $i$-th items [m, m$^2$, m$^3$, t, kg, pcs/m$^2$]; and $C_i$ is the conversion of the amount of the $i$-th item in VAC in m$^3$/Qi specific unit.

The apparent volume of packaging waste is expressed by the following equation:

$$VAP_i = VAC_i * C_{P_i} = Q_i * C_i * C_{P_i} \qquad \left[\text{m}^3/\text{m}^2\right] \tag{3}$$

where $VAP_i$ is the apparent volume of packaging waste of $i$-th items [m$^3$/m$^2$]; $VAC_i$ is the apparent constructed volume of $i$-th items [m$^3$/m$^2$]; $C_{P_i}$ is the transformation coefficient of packaging waste from $VAC_i$ to $VAP_i$ [dimensionless]; $Q_i$ is the quantity of $i$-th items [m, m$^2$, m$^3$, t, kg, pcs/m$^2$]; and $C_i$ is the conversion of the amount of the $i$-th item in VAC in m$^3$/Qi specific unit.

The estimated volume of the selected waste (m$^3$) associated with debris and packaging waste is calculated as a summary of multiplying $VADeb_i$ and $VAP_i$ by the building surface (m$^2$).

Further, the demolition of buildings is associated with the generation of demolished waste (Equation (4)).

$$VADem_i = VAC_i * C_{Dem_i} = Q_i * C_i * C_{Dem_i} \qquad \left[\text{m}^3/\text{m}^2\right] \tag{4}$$

where $VADem_i$ is the apparent volume of demolished waste of of $i$-th items [m$^3$/m$^2$]; $VAC_i$ is the apparent constructed volume of $i$-th items [m$^3$/m$^2$]; $C_{Dem_i}$ is the transformation coefficient of demolished waste [dimensionless]; $Q_i$ is the quantity of i-th items [m, m$^2$, m$^3$, t, kg, pcs/m$^2$]; and $C_i$ is the conversion of the amount of the i-th item in VAC in m$^3$/Qi specific unit.

The estimated volume of the selected waste (m$^3$) associated with demolished waste is calculated as a summary of multiplying $VADem_i$ by the building surface (m$^2$).

The transformation coefficients ($C_{Deb_i}$, $C_{P_i}$ and $C_{Dem_i}$) of the construction waste originated from the selected construction activity and construction elements. The coefficients were determined through a study of 45 construction projects, a questionnaire survey, and structured interviews.

The construction projects consisted of construction and demolition work. The necessary data were obtained from the construction project documentation, where the designers are required to estimate the amount of CDW generated under the building permit procedure. An expert estimate of the volume of the waste generated was determined based on a formula (Equation (5)):

$$C_{Deb_{iDES}} = \sum\nolimits_{j=1}^{n} Q_{Deb_{iDESj}} : n \tag{5}$$

where $C_{Deb_{iDES}}$ is the coefficient of debris waste determined based on the project documentation of the designed building; $Q_{Deb_{iDESj}}$ is the quantity of the $i$-th debris waste of the $j$-th construction projects in the project documentation [m, m$^2$, m$^3$, t, kg, pcs]; and $n$ is the number of analyzed construction projects.

Subsequently, the proposed coefficients were verified during the construction and demolition work through data on the amount of waste actually produced according to the formula (Equation (6)):

$$C_{Deb_{iCONS}} = \sum_{j=1}^{n} Q_{Deb_{iCONSj}} : n \qquad (6)$$

where $C_{Deb_{iCONS}}$ is the coefficient of the debris waste determined based on the actual amount of waste produced during the construction of a building; $Q_{Deb_{iCONSj}}$ is the quantity of the *i*-th debris waste of the *j*-th construction projects during the construction processes [m, m$^2$, m$^3$, t, kg, pcs]; and *n* is the number of analyzed construction projects.

Two calculated values formed the closed interval <$C_{Deb_{iDES}}$; $C_{Deb_{iCONS}}$> for the generation of the amount of *i*-th construction waste, which was assessed by a questionnaire survey. The questionnaire survey focused on estimating the amount of waste generated from specific construction activities and elements of construction. A total of 102 experts in the field of waste management in construction companies determined the amount of generated construction waste for selected structures in the defined closed interval. These results were verified through a structured targeted interview, which verified the idea of determining the quantity of coefficients [83].

## 4. Results and Discussion

Nowadays, the challenges of the AEC industries are decarbonization and the resource depletion of buildings as a result of sustainable construction projects. Construction and demolition waste are generated in each phase of the construction life cycle. At the same time, the volume of the generated CDW can be influenced during each stage.

### 4.1. Circular (Sustainable) Construction Design

The construction design phase determines the choice of construction technologies and, at the same time, the selection of construction materials. The stakeholders (the investor and designer) can already at this phase decide on a sustainable construction design while respecting the necessary technical, technological, economic, and environmental parameters of the construction design [84]. One of the important environmental requirements of construction is waste management, or the amount of CDW generated. The aim of the EU initiative concerning circular construction design [84] is the support and prioritization of a building design that: (i) produces less CDW during its life cycle, (ii) allows for the reuse of building materials and products, and (iii) reduces both the negative impact on the environment and costs throughout the life cycle of the construction.

Circular (sustainable) construction design (in the first phase of a building's life cycle) needs three basic pieces of information about the generated CDW: (i) the type of CDW, (ii) the volume, and (iii) the disposal way (Figure 3). It is not difficult to determine the type of waste generated or the way it is treated. The type of CDW generated depends on the type of construction activities proposed, which are influenced by the construction design. The CDW type will be classified according to the European Waste Catalogue [81]. The way of CDW disposal is determined by the objectives of the legislative framework [32,39]. The crucial point of this information is the volume of waste generated.

### 4.2. Application of Proposed Methods

This section provides the application of the proposed method for the waste quantification of four selected types of construction and demolition waste. The proposed methodology was applied to quantify the CDW on a construction project (Figure 4) with the following characteristics:

- Type: apartment house;
- Number of floors: 2;
- Number of apartments on 1 floor: 2;
- Surface area per apartment: 150 m$^2$;
- Surface area of other places: 45 m$^2$;

- Surface area together: 690 m$^2$;
- Structure: concrete foundation strips, masonry vertical load-bearing structures, reinforced concrete ceiling plate, horizontal roof.

A new construction project (apartment house) produces debris waste and packaging waste. At the same time, all four of the most significant waste streams (excavated soil, concrete, reinforced concrete, and masonry) are generated.

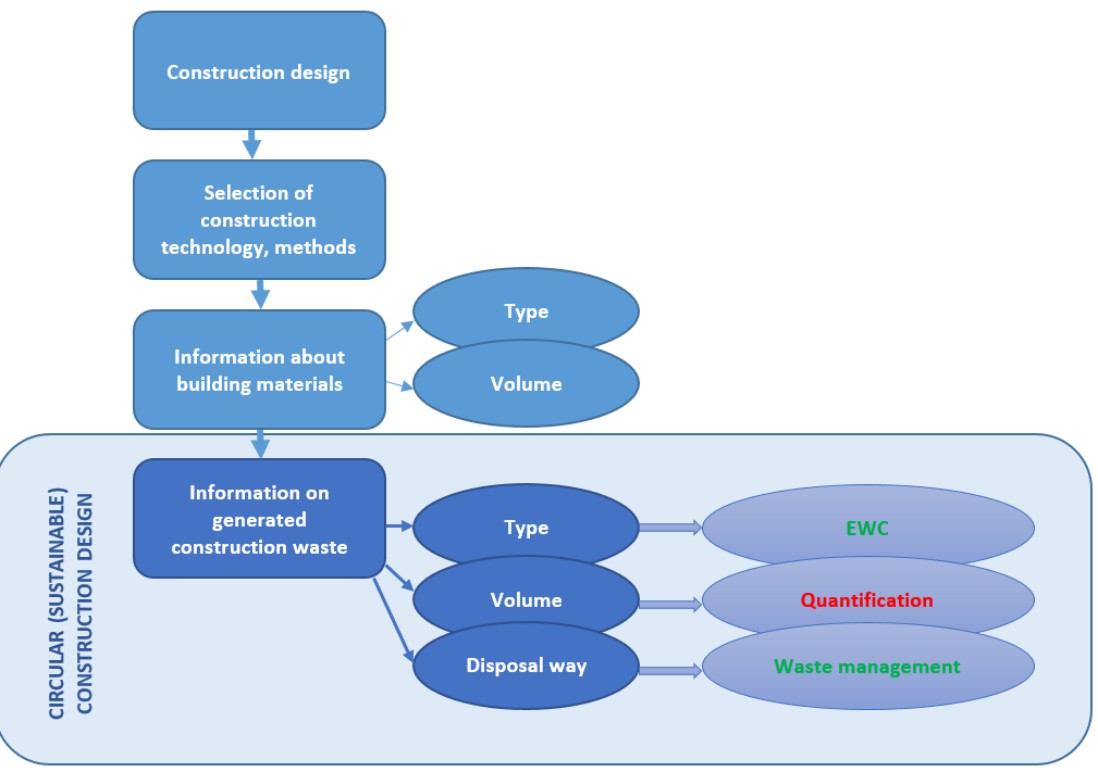

**Figure 3.** Circular (sustainable) construction design—first phase of construction life cycle.

(**a**)

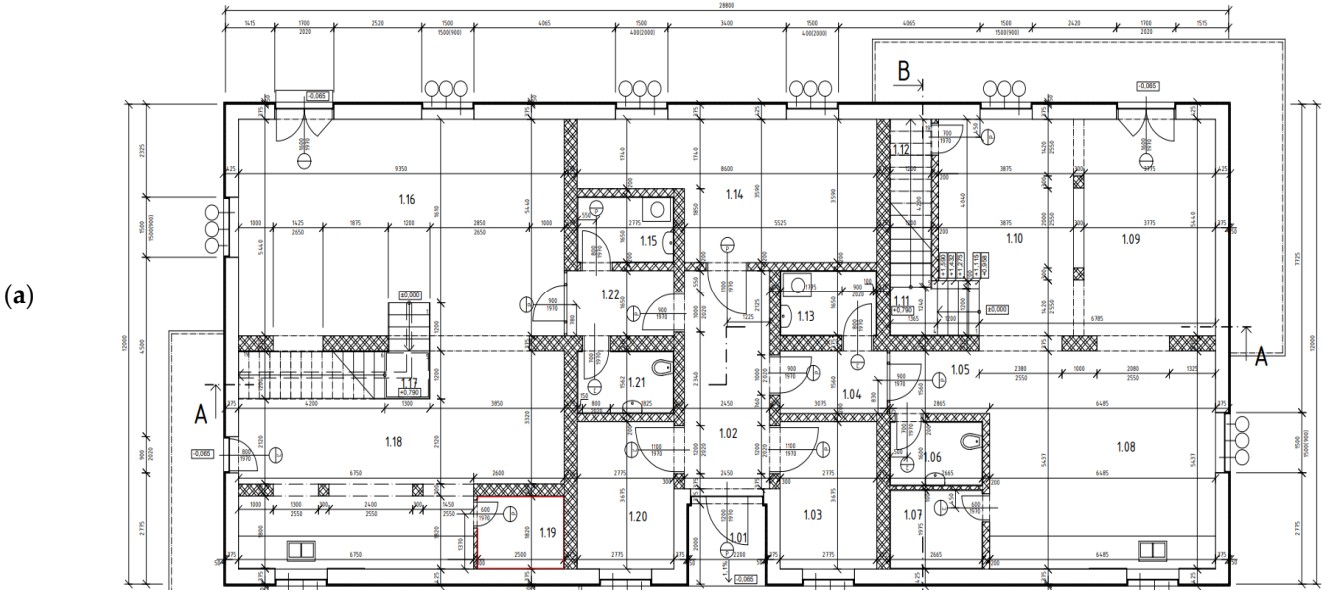

**Figure 4.** *Cont.*

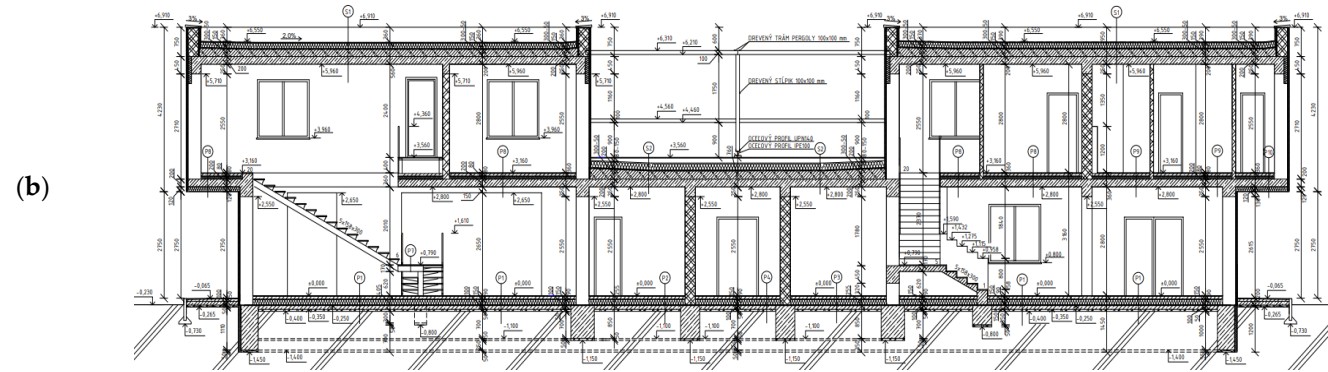

**Figure 4.** Construction project—apartment house: (**a**) floor plan; (**b**) section.

Table 4 summarizes the quantification of CDW by applying the proposed method. The calculation is explicit. The thickness of the excavated soil is 300 mm; therefore, the value of $Q_i$ is $0.30 \text{ m}^3/\text{m}^2$. The soil from the excavation is already defined in volume units, as cubic meters of loose soil, and then $C_i$ is 1.00. The total volume of the excavated soil is removed, which means that the transformation coefficient of the debris waste is 1.000. The excavated soil does not need to be packaged, and, therefore, the transformation coefficient of the packaging waste is 0.000. The same calculation procedure was performed for all types of waste. This calculation determined that the most significant volume of waste originates from the excavated soil. The volume of CDW in the masonry work is also not negligible; however, this waste consists mainly of packaging material.

**Table 4.** Quantification of CDW for construction works.

| CDW | | $Q_i$ | $C_i$ | $C_{DEBi}$ | $C_{Pi}$ | $VAC_i$ | $VA_{DEBi}$ | $VA_{Pi}$ | $\text{m}^3$ Waste/$\text{m}^2$ | $\text{m}^3$ Waste TOTAL (690 $\text{m}^2$) |
|---|---|---|---|---|---|---|---|---|---|---|
| Type | UM | | | | | | | | | |
| Transported excavated soil | $\text{m}^3$ | 0.30 | 1.000 | 1.000 | 0.000 | 0.300 | 0.300 | 0.000 | 0.300 | 207.0 |
| Concrete slabs | $\text{m}^2$ | 0.65 | 0.250 | 0.045 | 0.000 | 0.1625 | 0.007 | 0.000 | 0.007 | 5.046 |
| Masonry th. 375 mm | $\text{m}^2$ | 0.84 | 0.375 | 0.040 | 0.100 | 0.315 | 0.0126 | 0.0315 | 0.0441 | 3.,429 |
| Masonry th. 200 mm | $\text{m}^2$ | 0.92 | 0.200 | 0.050 | 0.100 | 0.184 | 0.0092 | 0.0184 | 0.0276 | 19.044 |

A similar calculation procedure was implemented to quantify the construction waste for demolition works. Only the demolished waste enters the calculation. Debris waste or packaging waste is not generated during demolition work. The calculation is again given for four selected types of construction waste (Table 5). The next item of the calculation will be detailed for a better understanding. The exterior load-bearing walls are of a thickness of 375 mm. The value of $C_i$ in this construction as cubic meters is 0.375 (Table 5). The transformation coefficient of the demolished waste for exterior masonry is 1.200 because the exterior walls are also demolished with layers of surface finishing, which increases the original volume of the construction elements by 20%. This calculation confirms an increase in the amount of waste in building elements, including surface treatments that were not removed before demolition work.

**Table 5.** Quantification of CDW for demolition works.

| CDW | | $Q_i$ | $C_i$ | $C_{DEMi}$ | $VAC_i$ | $VA_{DEMi}$ | $\text{m}^3$ Waste/$\text{m}^2$ | $\text{m}^3$ Waste TOTAL (690 $\text{m}^2$) |
|---|---|---|---|---|---|---|---|---|
| Type | UM | | | | | | | |
| Transported excavated soil | $\text{m}^3$ | 0.30 | 1.000 | 0.000 | 0.000 | 0.000 | 0.000 | 0.000 |
| Concrete slabs | $\text{m}^2$ | 0.65 | 0.250 | 1.150 | 0.1625 | 0.187 | 0.187 | 128.944 |
| Masonry th. 375 mm | $\text{m}^2$ | 0.84 | 0.375 | 1.200 | 0.315 | 0.378 | 0.378 | 260.82 |
| Masonry th. 200 mm | $\text{m}^2$ | 0.92 | 0.200 | 1.200 | 0.184 | 0.2208 | 0.2208 | 152.352 |

### 4.3. Practical Contribution

Information about the already generated CDW in the first phase of the construction life cycle can have a significant effect on the practical use of our methodology.

The proposed methodology can be classified as the generation rate calculation method, specifically as an area-based calculation. The paper's authors have identified two similar surveys according to area-based calculations in China [67] and Spain [85]. The methodology developed in Spain is focused on new construction activities, renovation, and demolition works. The total surface area was used as a multiplier. A similar method was used in research in China. Neither of the identified and studied research papers are from the area of Central Europe; therefore they are specific to the used building materials, technologies, and methods. Thus, the calculated values cannot be compared. Instead, use of the area-based calculation is their common feature.

The circular economy requires the CE's according principles of the 3Rs, the reduction, reusing, and recycling of the CDW generated in the construction life cycle. At present, there is a need to save raw materials, which are significantly consumed by the AEC industries. The initial information on waste generation, type, and quantity allows for a more sustainable approach to waste management. An economic analysis of CDW disposal [30] (including waste disposal costs and transport costs) has expanded the dimension of the decision-making process. The generated CDW, which enters the construction circular design, may contain information about the type, quantity, and way of waste disposal and, moreover, may be supplemented by economic values. Stakeholders (especially the investor, designer, and contractor) will be able to decide how to handle the CDW. Construction and demolition waste can enter the supply chain, for example through recycling, which is often more economical. The most unacceptable way of recovering CDW is landfilling, due to the properties of construction waste as a consequence of its reuse or recycling. Through a closed-loop, construction waste can be returned back into circulation, thus promoting and highlighting the benefits of the circular economy.

An audit of construction and demolition waste (CDWA) should be an integral part of each construction project, mainly demolition and renovation. A valid and proper CDWA must be processed by a qualified expert (authorized auditor). Authors of the study [80] determined that the mandatory pre-demolition and renovation audit are the most promising measures that impact the environmental and socio-economic fields of CDW management. The CDWA is not specified by a common framework across the EU. Compliance with the document is not mandatory, but its adoption will significantly increase the rate of construction waste recovery, thereby increasing the volume of CDW that returns to the supply chain. The processing of a CDWA has proven to be an effective tool in reducing the development of CDW, as an effective tool for proper waste management. A CDWA is required by legislative requirements, e.g., in France, the Netherlands, and the Scandinavian countries. This also reflects their excellent results in waste management. Alternatively, there are countries where this requirement does not exist, e.g., Slovakia, Greece, and Romania. Again, this is also reflected in these countries' waste management performances. The implementation of the CDWA should be part of project documentation, at least for reconstruction or demolition work. The favorable situation would be the implementation of the CDWA as part of the project documentation for all buildings (including new buildings). A well-processed CDWA provides the investor and contractor with valuable information on the future generation of demolition waste regarding the possibility of its separation and further processing for the purposes of CE application. The critical point of the CDWA is to determine the amount of waste generated. The proposed methods can also supplement this information for a properly executed CDWA. A well-processed CDWA will help the contractor to identify the quantity and quality of the generated waste at the beginning of construction, so that he can better choose the method for CDW management. At the same time, it is necessary for him to prioritize the separation of waste at the place of its origin, i.e., at the construction site. This can achieve the EU's waste management objectives, which are included in the Waste Framework Directive 2008/98/EC [32].

Building Information Modeling (BIM) is a tool that has a positive impact on increasing digital skills. In the context of the AEC industry, BIM means the process of delivering a project using well-structured digital information [86,87]. Data relating to waste management can be managed in the sixth dimension of BIM: sustainability. The sixth dimension of BIM is mainly used to assess energy efficiency during the design phase and the use phase. Efficient data collection will allow for a better understanding of the building's performance and define a strategy for optimization of the construction and demolition streams. Part of sustainability is also the area of waste management. Thus, it is appropriate to supplement the information model of the building with non-graphic information: the amount of waste, the type of waste, and the method for waste recovery. Again, BIM only offers another way of applying a CE in the first phase of the construction life cycle in order to use more recyclable building materials and more acceptable building practices, such as the selection of and preference for off-site construction methods that significantly reduce the waste of building materials, failures, and waste from the transport of building materials. BIM is one of the many tools, which can use the proposed CDW quantification methodology. At the same time, it supports the digitization of construction, which can increase its productivity from the point of view of waste management. The digitization of construction projects is one of the goals of the philosophy of Industry 4.0, such as Construction 4.0.

In addition, all of the proposed practical contributions are in line with the document from the United Nations "The 2030 Agenda for sustainable development", specifically: sustainable cities and communities; responsible consumption and production; and industry, innovation, and infrastructure [88].

*4.4. Limitation of Research*

Architecture, Engineering, and Construction are specific industries. Each building is unique. Each "production" area (construction site) is different. The buildings differ from each other in technical, technological, spatial, and material parameters. This paper provides examples only for selected construction and demolition waste and presents the gist of the proposal. Therefore, the proposed method is generally applicable. It will be necessary to use an expert estimate provided by experts in waste management in construction to quantify the CDW of specific or composite building materials.

The submitted research provides a method for the quantifying of CDW that generates input for optimizing the construction waste stream or other sophisticated waste management supported by ICT tools, such as building information modelling, a geographic information system, radio frequency identification, big data analytics, or the internet of things. The correct quantification of CDW, supported by an appropriate ICT tool, can increase the CE rate in construction and return CDW to the supply chain.

A life cycle assessment (LCA) is used for the quantitative and qualitative assessment of a particular product or material's environmental impact, energy consumption, and raw material consumption during the whole lifecycle of a product or material. Environmental impact assessment begins from a raw material's extraction to the landfill, representing the end of the life cycle [89]. In the case of AEC, the product presents as a building, and the material presents as construction material. The CDW have an exceptional property: it is possible to reuse them, thus ensuring their circularity [90]. According to Devaki and Shanmugapriya, the present is necessary to focus on sustainable waste management, which requires innovative approaches. These sophisticated approaches do not just address waste management. They are focused on processes and embody adaptability. The LCA of construction and demolition waste management dealt with many aspects, such as CDW collection, transport, recycling, or disposal, as well as the environmental point of view. The proposed methodology for CDW quantification and the determination of waste quantification can provide inputs for a detailed LCA of construction and demolition waste management, which is the subject of further research for the authors and the scientific community.

## 5. Conclusions

The construction sector is a sector with a high environmental burden, mainly due to the production of large amounts of waste. The construction industry consumes about 50% of total energy, 30% water, and 50% raw material. At the same time, it is responsible for generating more than a third of the total amount of waste [78,80]. Responsible waste management is an integral part of sustainable construction processes based on managing the waste stream on-site, waste minimization, waste registry, waste disposal (reusing, recycling, and reducing) on a construction site, and the policy and regulation in this field.

The effort to reuse waste gives hope for processing and recovery in a circular economy. Therefore, the need to address the issue is of great importance. The contribution of research results and the focus on quantification provides the essential information for practice and the effort to already help reduce waste significantly in the circular design of a building.

Research shows that effective waste management has a high potential for waste recovery and thus contributes to a friendly environmental approach in the construction sector. The research pointed out the importance of reducing waste according to the circular economy principles. The 3Rs approach is in the context of sustainability. It also has a deeper dimension in the perception of economic sustainability. Looking at this issue through economic sustainability means the search for solutions that will bring economic solutions. From this point of view, the quantification of waste and its use rate in the circular economy and circular design context is of great importance. This is also economically costly, saving material costs and waste disposal. This means that it has a double effect, and the principle of economic sustainability is respected [79,81].

The main principle of the circular economy in this issue is that CDW is thus returning to circulation. As a result, the supply chain is constantly replenished, saving minerals, which is economical and largely ecological. These approaches have been shown to lead to the fulfilment of non-environmental goals, which are also linked to economic ones, which deepen the essence of economic sustainability. This is important to maintain throughout the construction life cycle. From this point of view, the results show that the sustainability of buildings must be addressed in the first phase of the construction life cycle, i.e., in its design.

It is during the decision-making process and communication between the investor and the designer that it is necessary to try to choose more environmentally friendly variants of construction projects. One of these processes is the selection of more environmentally friendly materials. That is, building materials that have a higher degree of recyclability and reusability. These aspects will significantly help to achieve environmentally friendly solutions. With this goal in mind, the importance of this research is again demonstrated, as it is crucial to quantify the amount of waste at the design stage and other stages, especially during demolition work and the like. Quantification and calculations at this stage can help select materials with an additional focus on economic and non-environmental sustainability. Any precise method for determining the generation of CDW has not been developed at our national level, nor even at the level of Central European countries; however, comparable construction technologies, processes, and materials are used in these countries, which are required by comparable technical standards. Further, the construction conditions are almost identical—climate, altitude, etc. Therefore, we can assume that the proposed procedure for CDW quantifying is applicable.

The essential benefit for the industry is thus the proposed quantification of CDW, which directly contributes to all the benefits described above. In addition, the quantification proposal contributes to better results in the context of sustainability in construction. Likewise, shortening the supply chain, reducing the burden, and reusing materials and resources with the help of effective waste management is beneficial for the entire construction industry.

The proposed methodology for the quantification of construction and demolition waste considers the national specifics of the conditions of the Slovak construction industry, i.e., the used building materials, construction systems of the proposed buildings, the possibilities of

waste management, etc. At the same time, it can be used for new construction projects and reconstructions or demolitions of buildings. The application of the proposed methodology to construction practice can contribute to fulfilling waste management goals at the national and global levels.

**Author Contributions:** The individual contributions of authors were as follows: M.S. writing—original draft preparation, conceptualization, literature review, methodology, data processing; P.M.: writing—review and editing, supervising; T.M. and M.Š.: literature review, methodology. All authors have read and agreed to the published version of the manuscript.

**Funding:** This work was supported by the Slovak Research and Development Agency under the contract no. APVV-17-0549. This paper presents partial research results of project KEGA 009TUKE-4/2022 "An interactive tool for designing a safe construction site in an immersive virtual reality" and project VEGA, grant number 1/0336/22.

**Institutional Review Board Statement:** Not applicable.

**Informed Consent Statement:** The informed consent was obtained form all subject involved in the study.

**Data Availability Statement:** Not applicable.

**Conflicts of Interest:** The authors declare no conflict of interest.

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
