# Peer review of "Waste Management in a Sustainable Circular Economy as a Part of Design of Construction"

_applsci, doi:10.3390/app12094553_

Round 1

Reviewer 1 Report

The authors presented a study on Waste management in the sustainable circular economy as a part of design of construction.

The authors presented important issues of quantify construction and demolition waste in the construction design phase in the circular economy context.

The authors raise the issues Life-cycle of the building is divided into four main phases: (i) 126 design phase, (ii) manufacturing and construction phase, (iii) operation and maintenance 127 phase (use) and (iv) end of life (demolition). It's hard to find the LCA analysis and its results in the whole manuscript.

The authors presented process maps and selected inventory tables, but they are not suitable for LCA analyzes. I suggest changing the test method or carrying out a full LCA analysis with the use of software such as Opel LCA or SimaPro. In the current situation, we cannot talk about materials management and we cannot talk about LCA analysis at all.

Literature sources are cited in the conclusions - this should not be done. Conclusions should include the most important information from the conducted analysis - own research.

Author Response

Comments and Suggestions for Authors

The authors presented a study on Waste management in the sustainable circular economy as a part of design of construction.

The authors presented important issues of quantify construction and demolition waste in the construction design phase in the circular economy context.

The authors raise the issues Life-cycle of the building is divided into four main phases: (i) 126 design phase, (ii) manufacturing and construction phase, (iii) operation and maintenance 127 phase (use) and (iv) end of life (demolition). It's hard to find the LCA analysis and its results in the whole manuscript.

The authors presented process maps and selected inventory tables, but they are not suitable for LCA analyzes. I suggest changing the test method or carrying out a full LCA analysis with the use of software such as Opel LCA or SimaPro. In the current situation, we cannot talk about materials management and we cannot talk about LCA analysis at all.

partially accepted, corrected:

Thank you for your valuable advice and comments. I understand this recommendation, but the paper did not focus on LCA. The aim of the paper was to propose a quantification of construction and demolition waste in the construction design phase in the context of the circular economy. The quantification of CDW was based on the determination of coefficients for debris waste, packaging waste and demolition waste. The aim of the paper was not the LCA of building. This information only served as a knowledge base for research.

The issue of LCA and CDW management has been added to subchapter 4.4. Limitation of research:

A life cycle assessment (LCA) is used for quantitative and qualitative assessment of environmental impact of particular product or material, its energy consumption, consumption of raw material during whole lifecycle of product or material. The assessment of environmental impact begins from raw material extraction to landfilling, which represents the end of the life cycle [89]. In case of AEC, the product presents a building and material presents construction material. The CDW have an exceptional property. It is possible to reuse them, thus ensuring their circularity [90]. According Devaki and Shanmugapriya, it the present is necessary to focus on sustainable waste management which requires and innovative approaches. These sophisticated approaches do not just address waste management. They are focused on processes and embodies adaptability. LCA of construction and demolition waste management dealt with many aspects, such as CDW collection, transport, recycling or disposal and its environmental point of view. The proposed methodology of CDW quantification and determination of waste quantification can provide inputs for detailed LCA of construction and demolition waste management which is the subject of further research for the authors and the scientific community.

Literature sources are cited in the conclusions - this should not be done. Conclusions should include the most important information from the conducted analysis - own research.

accepted, corrected: - Conclusion is supplemented by the following text:

Any precise method of determining the generation of CDW has not been developed in our national level, moreover, not even at the level of Central European countries. In these countries, comparable construction technologies, processes and materials are used, which are required by comparable technical standards. Also, the conditions of construction are almost identical - climate, altitude, etc. Therefore, we can assume that the proposed procedure of CDW quantifying is applicable.

Reviewer 2 Report

This study investigated Waste management in the sustainable circular economy as apart of design of construction with some interesting results.

  1. Recycling and reuse of construction and demolition waste has been practicing in the construction sector for many years. The significant findings of the manuscript should be addressed and elaborated.
  2. Page 7, lines 280-290, regarding questionnaire survey, it might need to be elaborated. The detail information might need to be provided for determination of these transformation coefficients.
  3. The authors should compare their findings with other people’s works in literature.
  4. Revision of abstract and conclusions would be required.

Author Response

This study investigated Waste management in the sustainable circular economy as apart of design of construction with some interesting results.

  1. Recycling and reuse of construction and demolition waste has been practicing in the construction sector for many years. The significant findings of the manuscript should be addressed and elaborated.

accepted, corrected:

  • in subchapter 2.1. Construction and demolition waste:

In generally, an integral part of CDW issues is CDW management which consist of waste generation, waste collection, waste storage, waste processing and waste disposal [27, 28, 29]. Construction and demolition wastes are generated throughout the life cy-cle of building. We know four basic groups of CDW depending on which phase of the construction life cycle waste is generated: wastes from the operation of construction site facilities; wastes from the construction activities containing also preparatory, ancillary and transport of processes; waste from the use of the construction; wastes from the renovation, modernization and demolition of buildings [30, 31]. The waste management is determined by the waste management hierarchy [32]. CDW disposal is divided into five levels: prevention, re-use, recycling, recovery, disposal. The waste management hierarchy sets out the order of priority of the best environmental choice in the field of waste policy overall. The waste prevention is the most efficient and sustainable strategy for the use of natural resources. Waste reuse is a process that involves separating part of the waste suitable for further direct reuse. This waste may be used for the same activity from which it was generated or in another activity, without changing the properties or form of this waste. Increased attention needs to be paid to this method of waste disposal, especially in construction, given that there is a high potential for applying this principle. An example of this procedure in construction is the reuse of brick products, roofing, etc. The recycling presents a process of reusing previously used materials and products, prevents wastage of resources, reduces consumption of natural materials, reduces the amount of waste stored and reduces energy consumption, thus contributing to the reduction of greenhouse gas emissions against the use of primary materials. The recovery and disposal represent the least appropriate ways of CDW hierarchy from point of the sustainability, e.g. landfilling has a negative impact on environment (water pollution, greenhouse gas emission, global warming etc) [27, 28, 29, 43]. The waste management also includes the collection, storage and transport of CDW to the site if its processing. The waste processing includes at least pre-sorting, crushing or grinding, magnetic separation of metal elements and sorting.

  • in chapter 5. Conclusion:

Any precise method of determining the generation of CDW has not been developed in our national level, moreover, not even at the level of Central European countries. In these countries, comparable construction technologies, processes and materials are used, which are required by comparable technical standards. Also, the conditions of construction are almost identical - climate, altitude, etc. Therefore, we can assume that the proposed procedure of CDW quantifying is applicable.

2. Page 7, lines 280-290, regarding questionnaire survey, it might need to be elaborated. The detail information might need to be provided for determination of these transformation coefficients.

accepted, corrected:

Construction projects consisted of construction and demolition work. Necessary data were obtained from the construction project documentation, where the designers are required to estimate the amount of CDW generated under the building permit procedure. An expert estimate of volume of waste generated was determined on the basis of a formula (Equation 5):

                                                                                               (5)

where:         - coefficient of debris waste determined on the basis of project documentation of the designed building

 - quantity of i-th debris waste of j-th construction projects in project documentation [m, m2, m3, t, kg, pcs]

                             n – number of analyzed construction projects

 Subsequently, the proposed coefficients were verified during the construction work, resp. demolition work through data on the amount of waste actually produced according formula (Equation 6):

                                                                                             (6)

where:         - coefficient of debris waste determined on the basis of actual amount of waste produced during construction of building

 - quantity of i-th debris waste of j-th construction projects during construction processes [m, m2, m3, t, kg, pcs]

                             n – number of analyzed construction projects

Two calculated values formed the closed interval <; > of generation of the amount of i-th construction waste, which was assessed by questionnaire survey.

3. The authors should compare their findings with other people’s works in literature.

accepted, corrected:

Authors of paper have identified two similar surveys according to area-based calculation in China [67] and Spain [84]. The methodology developed in Spain is focused on new construction activities, renovation and demolition works. The total surface area was used as a multiplier. A similar method was used in research in China. Both identified and studied researches are not from the area of Central Europe, so they are specific to the used building materials, technologies and methods. Thus, the calculated values cannot be compared. On the other hand, the idea of use the area-based calculation is their common feature.

4. Revision of abstract and conclusions would be required.

accepted, corrected:

  • Abstract:

The lack of information about the quantity of CDW in design phase of building needed for sustainable design of construction was identified as a research gap.

The proposed method is based on generation rate calculation method.

The main contributions of the paper were identified in decision making process of sustainable building design, in the audit of CDW management, and in building information modelling as a support tool of CDW management.

  • Conclusion:

Any precise method of determining the generation of CDW has not been developed in our national level, moreover, not even at the level of Central European countries. In these countries, comparable construction technologies, processes and materials are used, which are required by comparable technical standards. Also, the conditions of construction are almost identical - climate, altitude, etc. Therefore, we can assume that the proposed procedure of CDW quantifying is applicable.

Reviewer 3 Report

the authors should address some issues that are in this study.

  • The abstract is very unclear; please revise it. It should be included the gaps, contributions, ….
  • Please revise the keywords; and they are very generic; in addition, the “keyword” should be removed after “keywords:”.
  • In line 37, “authors” should be removed from the names of the authors.
  • “The current understanding of CE lies in the idea of closed-loop.” The authors mentioned the closed-loop suddenly. It is better to explain more about it.
  • A circular economy should be defined by using wide acceptable references in the introduction.
  • The authors should highlight the research gaps in the introduction and explain the contribution of this study (both practical and to knowledge)
  • The authors didn’t explain some phases of the construction and demolition waste in section 2.1; please see:” Improving construction and demolition waste collection service in an urban area using a simheuristic approach: A case study in Sydney, Australia” and “A Synthesis of Express Analytic Hierarchy Process (EAHP) and Partial Least Squares-Structural Equations Modeling (PLS-SEM) for Sustainable Construction and Demolition Waste Management Assessment: The Case of Malaysia”.
  • Figure 1 is very unclear how the authors inferred this figure. It should be clearly discussed.
  • Section 2.3 is one of the most important sections of this paper, but the authors failed to review the literature critically. The authors should present all findings from the literature in a table. Summarising factors in this study is very important.
  • Figure 2 is a bit unclear. It is better to present this figure as a flow chart.
  • A further explanation should be included in section 4.1.

Author Response

the authors should address some issues that are in this study.

  • The abstract is very unclear; please revise it. It should be included the gaps, contributions, ….

accepted, corrected:

The lack of information about the quantity of CDW in design phase of building needed for sustainable design of construction was identified as a research gap.

The proposed method is based on generation rate calculation method.

The main contributions of the paper were identified in decision making process of sustainable building design, in the audit of CDW management, and in building information modelling as a support tool of CDW management.

  • Please revise the keywords; and they are very generic; in addition, the “keyword” should be removed after “keywords:”.

accepted, corrected:

waste management; sustainable circular economy; construction and demolition waste; quantifying of construction and demolition waste; construction projects

  • In line 37, “authors” should be removed from the names of the authors.

accepted, corrected:

Stahel and Reday [8] determined the features of the circular economy within the industrial economy.

  • “The current understanding of CE lies in the idea of closed-loop.” The authors mentioned the closed-loop suddenly. It is better to explain more about it.

accepted, corrected:

CE uses the idea of closed-loop in supply chains. According Wang and Wang [17], closed-loop “involves the movement of the products from the upstream suppliers to the downstream customers and the flow of used ones back to the remanufacturers, combine the forward logistics with the reverse logistics”. Closed-loop allows materials or wastes that are not used to be returned to the supply chain as recycled raw materials.

  • A circular economy should be defined by using wide acceptable references in the introduction.

accepted, corrected:

The most used definition of CE presented Ellen MacArthur Foundation, where CE defined as “an industrial economy that is restorative or regenerative by intention and design” [9]. CE from a supply chain point of view represents a closed loop material flow [10]. CE from point of utility value Webster [11] defined “the circular economy is one that is restorative by design, and which aims to keep products, components and materials at their highest utility and value, at all times”. The circular model is also intended to ensure a healthy environment. Profit in this system is based on the efficient use of natural resources through efficient use. And not only materials, but mainly products or their components. This significantly minimizes waste and costs for input materials and energy required for the production of new products.

  • The authors should highlight the research gaps in the introduction and explain the contribution of this study (both practical and to knowledge)

accepted, corrected: The research gap and contribution of this study are explained in chapter 3.

The research aim was set based on a detailed literature review. The literature review focused on four main research areas: circular economy, sustainability and supply chain; construction and demolition waste; life-cycle of building and construction material; and quantifying of CDW. A research gap was found out by this detailed literature review. Determining the method of quantification of CDW represents a gap in research in regional level. There are no relevant data on this issue in Slovakia [77]; at the same time, there is a lack of a comprehensive information base on the amount of CDW, which would provide relevant data already during the construction design [30, 77]. The construction design is a crucial phase for decision-making process for increase of reduce, reuse, recycle rate in accordance with the target of European Union [32]. The proposal of the methodology for the quantification of CDW in the construction design phase in the context of the circular economy represents the main contribution of the study.

  • The authors didn’t explain some phases of the construction and demolition waste in section 2.1; please see:” Improving construction and demolition waste collection service in an urban area using a simheuristic approach: A case study in Sydney, Australia” and “A Synthesis of Express Analytic Hierarchy Process (EAHP) and Partial Least Squares-Structural Equations Modeling (PLS-SEM) for Sustainable Construction and Demolition Waste Management Assessment: The Case of Malaysia”.

accepted, corrected:

In generally, an integral part of CDW issues is CDW management which consist of waste generation, waste collection, waste storage, waste processing and waste disposal [27, 28, 29]. Construction and demolition wastes are generated throughout the life cy-cle of building. We know four basic groups of CDW depending on which phase of the construction life cycle waste is generated: wastes from the operation of construction site facilities; wastes from the construction activities containing also preparatory, ancillary and transport of processes; waste from the use of the construction; wastes from the renovation, modernization and demolition of buildings [30, 31]. The waste management is determined by the waste management hierarchy [32]. CDW disposal is divided into five levels: prevention, re-use, recycling, recovery, disposal. The waste management hierarchy sets out the order of priority of the best environmental choice in the field of waste policy overall. The waste prevention is the most efficient and sustainable strategy for the use of natural resources. Waste reuse is a process that involves separating part of the waste suitable for further direct reuse. This waste may be used for the same activity from which it was generated or in another activity, without changing the properties or form of this waste. Increased attention needs to be paid to this method of waste disposal, especially in construction, given that there is a high potential for applying this principle. An example of this procedure in construction is the reuse of brick products, roofing, etc. The recycling presents a process of reusing previously used materials and products, prevents wastage of resources, reduces consumption of natural materials, reduces the amount of waste stored and reduces energy consumption, thus contributing to the reduction of greenhouse gas emissions against the use of primary materials. The recovery and disposal represent the least appropriate ways of CDW hierarchy from point of the sustainability, e.g. landfilling has a negative impact on environment (water pollution, greenhouse gas emission, global warming etc) [27, 28, 29, 43]. The waste management also includes the collection, storage and transport of CDW to the site if its processing. The waste processing includes at least pre-sorting, crushing or grinding, magnetic separation of metal elements and sorting.

  • Figure 1 is very unclear how the authors inferred this figure. It should be clearly discussed.

accepted, corrected:

We currently know of many construction approaches that allow the construction of off-site, e.g. modern methods of construction, prefabrication, etc. [42]. Renovation or modernization of building may occur during its use phase. These activities require the supply of building materials, but also create CDW.

  • Section 2.3 is one of the most important sections of this paper, but the authors failed to review the literature critically. The authors should present all findings from the literature in a table. Summarising factors in this study is very important.

accepted, corrected:

CDW quantification methods can be classified according to construction activities and estimation levels (Table 1).

Table 1. Classification of CDW quantification methods according to construction activities and estimation levels.

CDW quantification method

Construction activity

Estimation level

Source

Construction of new buildings

Demolition of buildings

Civil and infrastructural works

Project level

Region level

DSVM

x

x

[60]

ISVM

x

x

x

[53]

ISVM

x

x

x

[61]

GRC – PC

x

x

x

[63]

GRC – PC

x

x

[64]

GRC – FVE

x

x

[64]

GRC – ABC

x

x

x

[65]

GRC – ABC

x

x

x

[66]

LAM – BLA

x

x

x

[53]

LAM – BLA

x

x

[56]

LAM – MLA

x

x

x

[67]

CSAM

x

x

[55]

CSAM

x

x

x

[65]

VMM

x

x

[64]

DSVM - direct site visit method; ISVM - indirect site visit method; GRC – PC - generation rate calculation method – per capita; GRC – FVE - generation rate calculation method – financial value extrapolation; GRC – ABC - generation rate calculation method – area-based calculation; LAM – BLA - lifetime analysis method – building lifetime analysis; LAM – MLA - lifetime analysis method – material lifetime analysis; CSAM – classification system accumulation method; VMM – variables modelling methods.

The accuracy and clarity of each method varies. Methods require different input data, which are not always available, resp. their relevance cannot be verified. Limitations of quantifying of CDW are mainly lack of actual data and lack of verification.

Site visit quantification methods are based on a visit to a specific construction site construction or demolition waste is generated [53, 60, 61]. A direct method determines the amount of waste based on spatial dimensions of place stockpiled CDW. An indirect method consists in counting the number of trucks for waste transport and volume of each container.

The generation rate calculation methods are the most used methods of CDW quantification. The principle of this methodology is to obtain the rate of waste generation (coefficient) per specific unit (e.g.: kg/m2, m3/m2) by different type of multiplier [63-66]. It has a general use. The critical point of using this method is the absence of suitable databases of rate calculation.

The lifetime analysis method is used for demolition waste quantification. Building lifetime analysis takes into account the age of the buildings, which often determines the need for demolition work [53, 56]. The national material flow databases are essential for the application of the material lifetime analysis method. Database determines the amount of purchased building material for a given region [67]. The lifetime analysis methods are not generally applicable to every construction project and region. These databases have been researched and designed by this methodology for the United States only.

The classification system accumulation method comes from the idea of generation rate calculation methods. It is focused on CDW quantification for a specific waste type and a specific building structure of element [55, 65].

The variables modelling methods consists of modelling various variables that arise during construction or demolition activities. These variables are e.g. construction time, construction site conditions, weather, material storage or labour experience. The importance and frequency of these variables affect the generation of construction waste [64].

  • Figure 2 is a bit unclear. It is better to present this figure as a flow chart.

accepted, corrected – figure 2 has not been replaced by flow chart, but was supplemented by directions of procedure and described in section 4.1

  • A further explanation should be included in section 4.1.

accepted, corrected:

The research aim was set based on a detailed literature review. The literature re-view focused on four main research areas: circular economy, sustainability and supply chain; construction and demolition waste; life-cycle of building and construction material; and quantifying of CDW. A research gap was found out by this detailed literature review. Determining the method of quantification of CDW represents a gap in re-search in regional level. There are no relevant data on this issue in Slovakia [66]; at the same time, there is a lack of a comprehensive information base on the amount of CDW, which would provide relevant data already during the construction design [31,66]. The construction design is a crucial phase for decision-making process for increase of re-duce, reuse, recycle rate in accordance with the target of European Union [29]. The proposal of the methodology for the quantification of CDW in the construction design phase in the context of the circular economy represents the main contribution of the study.

Round 2

Reviewer 1 Report

The authors have sufficiently corrected their typescript. Changes in flowcharts were checked. Autros cleared up the dubious issues. I recommend that the typescript be accepted for publication after checking the spelling and editing the text.

Reviewer 3 Report

The authors addressed all comments.